# Study protocol to establish a prospective cohort for the study of phenotypic clusters, progression pathways, and outcomes of frailty and dependence: The CohorFES

Natàlia Garcia-Giralt[1,2]*, Diana Ovejero[1,2], Jose Antonio Carnicero Carreño[1,3], Anna Ribes[2], Pedro Abizanda Soler[1,4], Jose Antonio Serra Rexach[1,5], Francisco José García García[1,6], Montse Rabassa[1,7], Leocadio Rodriguez Mañas[1,3], Àlex Sanchez Pla[1,8], Mariam El Assar de la Fuente[1,3], Carmen Maria Osuna Del pozo[5], Inmaculada Carmona[5], María Ángeles Caballero-Mora[9], Virginia Mazoteras Muñoz[9], Elisa Belen Cortes Zamora[4], Almudena Avendaño Céspedes[1,4], Bárbara Agud Andreu[10], Fernando Gómez Galera[10], Jade Soldado-Folgado[1,2], Maria Cristina Andrés Lacueva[1,7], Xavier Nogués[1,2]

1 CIBER on Frailty and Healthy Ageing (CIBERFES), ISCIII, Spain, 2 Hospital del Mar Research Institute, Barcelona, Spain, 3 Fundación para la Investigación Biomédica del Hospital Universitario de Getafe, Instituto de Investigación Sanitaria Hospital Universitario de Getafe (IISGetafe), Getafe, Spain, 4 Complejo Hospitalario Universitario de Albacete, SESCAM, Albacete, Spain, 5 Hospital General Universitario Gregorio Marañón Madrid, Madrid, Spain, 6 Complejo Hospitalario de Toledo SESCAM, Toledo, Spain, 7 Departament de Nutrició, Ciències de l'Alimentació i Gastronomia, Nutrition and Food Safety Research Institute (INSA), Facultat de Farmàcia i Ciències de l'Alimentació, Universitat de Barcelona (UB), Barcelona, Spain, 8 Departament de Genètica, Microbiología i Estadística, Universitat de Barcelona, Barcelona, Spain, 9 Servicio de Geriatría, Hospital General Universitario de Ciudad Real, SESCAM, Ciudad Real y Instituto de Investigación Sanitaria de Castilla la Mancha, IDISCAM, Toledo, Spain, 10 Laboratori de Referència de Catalunya, Hospital del Mar, Barcelona, Spain

* ngarcia@researchmar.net

## Abstract

Frailty has become a major challenge for health systems, but it also presents a window of opportunity to fight disability through preventive strategies focused on the detection and treatment of frailty across all care settings. However, no systematic strategies for screening and early detection are currently available in clinical practice. This project aims to identify clinical and biological phenotypic clusters that drive progression through the different stages of frailty, and to describe the underlying mechanisms of the trajectories leading to disability and potential treatment interventions. In addition, the Frailty Trait Scale 5 (FTS5) will be validated as a practical tool for implementation in both primary care and hospital settings. A prospective, population-based cohort (CohorFES) will be established for frailty phenotyping. A CIBERFES Biobank will also be created to store blood and urine samples from CohorFES participants for future research. Demographic and clinical history data, anthropometric measurements, the PREDIMED questionnaire, peripheral blood biochemical variables, and metabolomics data will be collected at baseline and annually until participants develop frailty. Cluster partition models (k-means and hierarchical

**Data availability statement:** Deidentified research data will be made publicly available when the study is completed and published. All data underlying the findings will be fully available without restriction in re3data.org at the time of publication.

**Funding:** The present study is funded by CIBER on Frailty and Healthy Ageing, by Francisco Soria Melguizo Fundation, and FIS (ISCIII) num. PI19/00033. Additional support was provided by AGAUR (2021 SGR 00043), and FEDER, EU. CIBER on Frailty and Healthy Ageing, Francisco Soria Melguizo Fundation, and AGAUR participate in personal finance initiatives as well as contractual financing arrangements. FIS (ISCIII) are involved in laboratory material acquirement for sample and data collection. FEDER supports Spanish grants. Funders had no role in study design, analysis, decision to publish, or preparation of the manuscript.

**Competing interests:** The authors have declared that no competing interests exist.

clustering) will be used to group individuals with similar deficits and characteristics (frailty phenotypes). Then, by using pre-established criteria (gap and silhouette), the proposed clustering solution (belonging to given clusters) will be evaluated. We will also assess, in a longitudinal manner, the emergence and accumulation of deficits over time, identifying subgroups with more rapid progression. The results will be used to define and compare clusters and progression trajectories. Finally, frailty phenotypes and patient clusters will be correlated with health outcomes such as healthcare utilization (primary and secondary care), hospital admissions, and mortality. Information on clinical and biological phenotypic clusters involved in the progression of frailty may help identify potential therapeutic targets to improve the management of these patients. In summary, from a research perspective, this project aims to improve our understanding of the interindividual variability in clinical trajectories that lead to frailty, dependence, and ultimately, death.

**Protocol Registration:** NCT06965972 (date 05/02/2025)

## Introduction

Frailty is one of the main challenges of the 21st century and a top priority for national and international organizations such as the WHO (World Health Organization) and the European Parliament. This has placed it as one of the top priorities on the European Commission's biomedical research agenda [1,2]. Frailty is characterized by a physiological state of increased vulnerability and decreased resilience to stressors (i.e., diseases, external agents, drug tolerability and toxicity) due to the combined effect of the aging process and some chronic diseases, leading to a final phase of dependency and disability, with a significant impact on quality of life, consumption of healthcare and social resources, hospitalization, and death [2].

The importance of frailty detection and management is well recognized, given its reversibility [3], the associated healthcare costs [4], and its clinical implications [5]. In contrast to the abundance of data from non-clinical settings, there is a lack of robust data in clinical settings, where frailty prevalence is highest and the risk of serious adverse consequences is greatest [6]. Therefore, there is an urgent need to improve the detection and diagnosis of frailty, its trajectories, and the determinants of these trajectories, depending on the patient's characteristics (whether or not associated with sarcopenia, cognitive impairment, or various chronic diseases).

While the different categories of frailty syndrome (robust, frail, pre-frail) are well defined and characterized epidemiologically [7,8], there is little data on the functional pathways between these diagnostic categories (and, notably, disability), especially in clinical cohorts. This is particularly striking given that one of the most relevant factors associated with poor frailty outcomes is hospitalization [9].

Another key issue is the implementation of an easy-to-use tool to identify frailty and related factors in all outpatients. The Frailty Trait Scale (FTS) has demonstrated

the ability to detect frailty in some clinical settings (Acute Care Geriatric Units, Geriatric outpatient clinics, and Primary Care) and has good predictive capacity for adverse outcomes (death, incident disability, decline in SPPB score, falls, and hospitalization) at 6, 12, and 18 months [10,11]. However, the full version of the FTS, composed of 12 items, takes about 15 minutes to complete, making it impractical in routine clinical settings where time is limited.

With this in mind, a shorter version comprising only 5 items (the so-called FTS5) was developed [12]. This version offers promising results, maintaining the sensitivity to detect small changes shown by the full FTS. The variables that make up the FTS5 (gait velocity, grip strength, BMI, PASE, and balance) can also be incorporated into electronic tools, such as the electronic frailty index (eFI) [13]. The use of such electronic tools has proven useful not only in hospital care but also in routine primary care practice. Furthermore, it facilitates the measurement of adverse outcomes, including falls, delirium, disability, care home admission, hospitalization, and mortality, as recently demonstrated [14].

In this regard, several clinical guidelines have been proposed for the early identification and prevention of frailty in the elderly [15–17]. These guidelines aim to address one of the major healthcare challenges: the growing aging population and the increasing incidence of chronic diseases. In particular, in 2022, the Frailty and Falls Prevention Working Group updated a "consensus document on the prevention of frailty in the elderly," approved by the Interterritorial Council of the Spanish National Health System. This document highlights the importance of early diagnosis and the need for interventions within the healthcare system, especially in primary care [15]. This aspect is particularly important because frailty is not an inevitable consequence of aging; instead, it represents a dynamic and potentially reversible condition, with early stages offering the greatest opportunity for recovery.Therefore, early detection and diagnosis of frailty are essential not only for quality of life but also for the efficient use of healthcare and social resources.

Our overarching goal is to identify critical subgroups of individuals at risk of progressing from robustness to pre-frailty and frailty, and from there to later stages, as well as the pathways mediating this trajectory in community-dwelling Spanish subjects.

Within this conceptual framework, and considering the limited data available in clinical settings regarding frailty diagnosis, trajectories, and prognosis, the main goal of this project is to establish a real-life, clinical, prospective cohort (CohorFES). This cohort aims to identify clinical and biological phenotypic clusters contributing to progression through the different stages of frailty and to elucidate the underlying mechanisms that ultimately lead to disability.

Additionally, the project will validate the Frailty Trait Scale 5 (FTS5) as a practical and easily applicable tool for use in both primary care and hospital settings.

Finally, a multicenter biobank will be created within the CIBERFES consortium to store blood and urine samples from CohorFES participants, ensuring their availability for future research initiatives.

## Materials and methods

### Study design and setting

We will stablish a multicentered cohort, the CohorFES, a prospective and observational study based on population. The study is registered in clinicaltrials.Gov as observational study: NCT06965972 (date 05/02/2025). Patients will be recruited from the start of the project and will be followed annually throughout the study.

Participants are women and men over 65 years old who attend outpatient clinics at participating centers, including the Hospital del Mar Research Institute in Barcelona, the Hospital General Universitario Gregorio Marañón in Madrid, the Complejo Hospitalario Universitario de Albacete in Albacete, the Hospital General Universitario de Ciudad Real in Ciudad Real, the Fundación del Hospital Nacional de Parapléjicos in Toledo, and the Hospital Universitario de Getafe in Madrid.

### Participant selection for cohort study

Inclusion criteria consisted of women and men older than 65 years who had provided signed written informed consent. Exclusion criteria included patients in a critical end-of-life situation or with a Barthel Index score below 60, patients who

had been hospitalized for any illness within the previous six months, and patients diagnosed with and currently receiving treatment for any type of cancer, with the exception of non-melanoma cutaneous cancer.

## Sampling procedure

Individuals who are visited at participating centers and meet the inclusion criteria are invited to participate in the study. Participants are enrolled consecutively after providing written informed consent. All individuals included in the CohorFES are also invited to participate in the Biobank, after informed consent was obtained. Serum, plasma, buffy coat, and urine samples will be collected at baseline (V0) and annually thereafter.

## Sample size

The sample size calculation for this study is based on index data from the Toledo Healthy Aging Study. Since the prevalence rates of frailty, as well as the progression of frailty syndrome, are expected to be higher in the clinical population than in the general population, these estimates can be considered conservative and therefore offer an additional safety margin to achieve sufficient statistical power.

Frailty rates are estimated from subjects who have a negative FTS5 that later develop frailty, which was 0.075 (7.5%). The ratio of exposed/ unexposed was 1008/196 (ratio of negatives vs. positives), that is 5.1. The minimum relative risk value (RR) to be detected is estimated taking as an interest variable "incident frailty" (stage transition marker) for which OR of the different estimates range between 1.78 and 5.4 and therefore we adopt a RR conservative to detect of 2. Finally, the loss of follow-up rates is estimated at 10%.

With these data, sample size was calculated using a Poisson distribution approach, appropriate for the analysis of event rates. The calculation was based on the expected incidence rate of the outcome, the anticipated rate ratio between groups, a two-sided significance level of 0.05, and a statistical power of 80%. Therefore, accepting an Type 1 error of 0.05 and a Type II error of 0.2 in a two-sided contrast, 215 subjects in the exposed group and 1075 in the unexposed group are needed to detect a factor of risk with a minimum relative risk value of 2.

## Study schedule

This is a prospective study with long-term follow-up, where patients will have a visit every 12 months for all study procedures. We anticipate continuing follow-up until 1,500 patients are included, with a minimum follow-up duration of one year per participant.

The recruitment period will run from January 2, 2021, to December 31, 2027. Data will be collected throughout the entire study period, which is expected to conclude in 2030. At that point, all data will be analyzed and the results will be reported.

The recruitment process comprises a preselection visit to assess inclusion and exclusion criteria and to inform the participant about the study, followed by a baseline visit during which all study variables will be collected and written informed consent will be obtained. Each participant will complete a minimum of three study visits. These will include the baseline visit (V0) and 2 years of follow-up (V1 at 1 year of follow-up and V2 at two years of follow-up). Follow-up visits will be conducted every 12 months until the end of the study; during these visits, the same clinical variables collected at baseline will be reassessed, and biological samples will be collected. A final visit may take place upon completion of the study, participant withdrawal, or the diagnosis of frailty or death.

## Variables

All variables and biological samples will be obtained at baseline and at all subsequent visits until the end of the study or the patient's withdrawal (Fig 1).

**Participant timeline: Schedule of enrollment, interventions, and assessments.**

| | Project PERIOD | | | | | |
| | Enrollment | | Follow-up | | | Close-out |
| TIMEPOINT | $-t_i$ to 0 | 0 | 1 year | 2 year | etc. | $t_x$ |
| ENROLLMENT: | | | | | | |
| Eligibility screen | X | | | | | |
| Informed consent | | X | | | | |
| ASSESSMENTS: | | | | | | |
| Frailty score | | X | X | X | X | X |
| Clinical data | | X | X | X | X | X |
| Anthropometric measurements | | X | X | X | X | X |
| Biochemical variables | | X | X | X | X | X |
| SAMPLE COLLECTION: | | | | | | |
| Blood samples | | X | X | X | X | X |
| Urine | | X | X | X | X | X |

**Fig 1. SPIRIT schedule enrollment.**

**Main variables.** The main variables are based on the evaluation of the frailty score:

- **Frailty Trait Scale 5 items (FTS5)**: 1.- walking speed test, 2.- grip strength, 3.- Physical Activity, 4- Body Mass Index (BMI), and 5.- progressive Romberg test. Each of the 5 domains scores from 0 (the lowest) to 10 (the highest), being 50 the highest score possible in this tool. Participants with a score higher than 25 were considered as frail [12].

- **Fried phenotype**: 1.-weight loss, 2.-exhaustion, 3.-weakness, slowness, and 4.-low physical activity. Frailty status was defined as robust or nonfrail (individuals who did not meet any criteria), prefrail (1 or 2 criteria met), and frail (≥3) [7].

- **Electronic Frailty index (eFI):** It is a validated instrument that operationalizes the cumulative deficit model of frailty, using routinely collected electronic health record data (36 items) to quantify frailty and stratify older adults according to their risk of adverse outcomes such as hospitalization, institutionalization, and mortality [13].

These three frailty parameters will be compared to each other to validate whether the FTS5 could be used as a practical and easy-to-apply tool in both primary care and hospital settings.

**Secondary variables.** For each patient, the following data is obtained from the clinical history and through questionnaires:

*Demographic data:* Age in years,date of birth, sex, living situation, and educational level

1. *Clinical history data*

   - Prevalent diagnosis

   - Loss of weight in the last year

- Geriatric Depression Scale: The Geriatric Depression Scale (GDS) is a self-report measure of depression in older adults. Users respond in a "Yes/No" format. The GDS was originally developed as a 30-item instrument. Since this version proved both time-consuming and difficult for some patients to complete, a 15-item version was developed. The shortened form (GDS-S) is comprised of 15 items chosen from the Geriatric Depression Scale-Long Form (GDS-L). These 15 items were chosen because of their high correlation with depressive symptoms.

- Barthel Index: Barthel Index is an ordinal scale used to measure performance in basic activities of daily living, i.e., feeding, bathing, grooming, dressing, bowel control, bladder control, toileting, chair transfer, ambulation and stair climbing. The index also indicates the need for assistance in care. It is a widely used measure of functional disability. The index was developed for use in rehabilitation patients with stroke and other neuromuscular or musculoskeletal disorders, but may also be used for oncology patients.

- Lawton-Brody Instrumental Activities of Daily Living Scale: is an instrument to assess independent living skills. It takes approximately 10–15 minutes to administer and contains 8 items that are rated with a summary score from 0 (low functioning) to 8 (high functioning).

- Pfeiffer test: Also known as Short Portable Mental Status Questionnaire (SPMSQ), it is a brief test of 10 questions used to assess a person's cognitive status, especially in older adults. It evaluates aspects such as memory, temporal and spatial orientation, and the ability to perform basic tasks.

- Predimed questionnaire: The adherence of participants to the Mediterranean diet will be assessed through the 14-item Mediterranean diet adherence screener (MEDAS) validated for the Spanish population in a phone interview with the participant [18]. It consists of two questions about eating habits, eight questions about the frequency consumption of typical foods of the Mediterranean diet, and four questions about the consumption of foods not recommended in this diet. Each question is scored with 0 (non-compliant) or 1 (compliant), and the total score (from 14 questions) ranged from 0 to 14, so a score of 14 points means maximum adherence [18].

*Anthropometric measurements* (in the case of the center has the facilities and densitometer):Total body dual-energy X-ray analysis using a DXA device for the measurement of bone mineral density (both overall and in regions of interest) and body composition (fat mass, muscle mass).

*Peripheral blood biochemical variables:* Leukocytes ml/mmc, Red blood cells ml/mmc, Hemoglobin g/dl, Hematocrit %, Red blood cell distribution width %, Platelets ml/mmc, Vitamin D ng/ml, TSH nmol/l, Glucose mg/dl, Glycated hemoglobin (diabetics only) %, Creatinine mg/dl, Sodium meq/dl, Potassium meq/dl, Calcium mg/dl, Phosphorus mg/dl, GPT u/l, GOT u/l, Phosphatase alkaline u/l, Total proteins g/dl, Albumin g/dl, Prealbumin mg/dl, Cholesterol mg/dl, Triglycerides mg/dl, HDL mg/dl, LDL mg/dl, and C-reactive protein mg/ml.

*Metabolomic analysis*

a) Exposome assay

A targeted quantitative metabolomics approach will be applied to analyze the plasma samples using liquid chromatography coupled with tandem mass spectrometry (LC-MS/MS), as described previously [19,20]. This method will employ analyte extraction and LC separation combined with selective mass-spectrometric detection using multiple reaction monitoring (MRM) pairs to identify and quantify metabolites.

The approach will allow for the comprehensive profiling of over 1100 metabolites, including dietary-derived metabolites (especially (poly)phenols and their microbiota-derived derivatives); other dietary markers such as non-(poly)phenolic compounds (e.g., glucosinolates, methylzanthines, alkaloids, amino acid derivatives such as peptides, S-allylcysteine, fatty acids, benzoxazinoids); environmental and lifestyle-related metabolites (such as pollutants, pharmaceuticals, nicotine and alcohol metabolites); and approximately 500 endogenous compounds involved in key metabolic pathways (e.g., amino

acids and derivatives, lipids, vitamins, biogenic amines, organic acids, and carbohydrates). The panel will also include metabolites from the tryptophan-kynurenine pathway, as well as and microbial-derived metabolites such as indoles, bile acids, and short-chain fatty acids and others.

Plasma samples will be stored at −80 °C until analysis. For extraction, 10 µL of internal standard solution will be added to 100 µL of plasma, followed by 500 µL of cold acetonitrile (−20 °C) containing 1.5 M formic acid and 10 mM ammonium formate in the plate. After vortexing and incubation at −20 °C to induce protein precipitation, the extracts will be collected using a Waters Positive Pressure-96 processor (Waters, Milford, MA, USA), dried under a stream of nitrogen gas, and reconstituted in 100 µL of water:acetonitrile (80:20, v/v) containing 0.1% formic acid (v/v) and external standards. The samples will then be vortexed and prepared for injection.

Mass spectrometric analysis will be performed on a Sciex 7500 QTrap® instrument coupled with an Agilent 1290 UHPLC system. A Luna Omega Polar C18 column will be used for chromatographic separation, with distinct different gradients applied for positive and negative ion modes. The instrument will operate in scheduled MRM (sMRM) mode to ensure high sensitivity and selectivity for all targeted metabolites.

b) Clinical biomarkers assay

Plasma samples will be prepared and analyzed using a combination of direct flow injection tandem mass spectrometry (DFI-MS/MS) and liquid chromatography tandem mass spectrometry (LC-MS/MS) (20).

This targeted clinical biomarker assay will aim to provide a robust and user-friendly platform for the discovery and quantification of metabolomic biomarkers. The method will allow the targeted identification and quantification of up to 651 metabolites from 18 different analyte groups, including amino acids and derivatives, biogenic amines, organic acids, nucleotide/nucleosides, lipids, acylcarnitines, and glycerophospholipids. The analytical workflow will combine analyte derivatization, extraction and high-throughput quantification using MRM in tandem mass spectrometry. For accurate quantification, the method will incorporate stable isotope-labelled internal standards, along with other compound-specific internal standards, across all metabolite groups.

*Healthcare resource use:* Number of Primary Care consultations, prescribed drugs (polypharmacy), contacts with the Secondary Care facilities during the study period.

Non-elective hospital admissions and number of admissions and length of stay during the study period.

*All-cause mortality.*

## Statistical analysis plan

Descriptive statistics will be used to summarize baseline characteristics of the study population. Continuous variables will be reported as means and standard deviations or medians and interquartile ranges, as appropriate, while categorical variables will be summarized using frequencies and percentages. All statistical analyses will be conducted using R software, and a two-sided significance level of 0.05 will be applied unless otherwise specified.

To identification of frailty phenotypes, unsupervised cluster partitioning methods will be applied to identify groups of individuals with similar patterns of deficits and clinical characteristics, thereby defining distinct frailty phenotypes. Specifically, k-means clustering and hierarchical clustering approaches will be used. Prior to clustering, variables will be standardized as appropriate to ensure comparability across measures. The optimal number of clusters and the quality of the clustering solutions will be evaluated using predefined validation criteria, including the gap statistic and silhouette index. These metrics will be used to assess cluster separation, cohesion, and stability, as well as the robustness of individual cluster assignments.

In a second step, a longitudinal follow-up will be conducted to record the onset and accumulation of health deficits throughout the study period, continuing until the end of follow-up, death, or loss to follow-up. To delineate distinct patterns of frailty progression over time, we will apply latent class analysis using mixed growth models or comparable latent

trajectory modelling approaches. These methods will allow to identify differential longitudinal trajectories characterized either by varying rates of deficit accumulation (capturing acceleration in frailty development) or by distinct combinations of specific deficits or phenotypic characteristics.The resulting models will classify participants into latent clusters representing heterogeneous frailty trajectories, enabling the identification of subgroups at higher risk of accelerated decline and facilitating targeted intervention strategies.

Finally, the identified frailty phenotypes and patient clusters will be examined in relation to clinically relevant health outcomes, including the utilization of health care services in both primary and secondary care settings, hospital admissions, biomarker profiles, and mortality. Appropriate regression models will be used depending on the outcome of interest, and results will be reported using effect estimates with corresponding 95% confidence intervals. Where applicable, analyses will account for follow-up time and relevant covariates. These analyses will enable the assessment of the prognostic and clinical relevance of the identified frailty phenotypes and trajectories.

### Data collection and management

We use a data collection platform (REDCAP) designed by Hospital Universitario de Getafe, accessible to all project researchers. Participant identification is encoded at the time of inclusion in REDCAP. The project's principal investigator (PI) can export REDCAP data to various applications (Excel, SPSS, etc.). The coordinating researchers will manage the anonymized data for project analyses and data confidentialy. Data will be validated by PI and statistical researchers.

### Potential bias

All centers involved in participant recruitment have a specific outpatient clinic focused on frailty and the management of older adults. These centers are geographically distributed throughout Spain, minimizing potential bias.

### Limitations

The main limitation of the project is the coordination of patient recruitment among all participant centers since the data homogeneity and sample size is a key factor for the establishment of the CohorFES.

### Ethical consideration

The study follows the national and international standards (Declaration of Helsinki and Tokyo) on ethical aspects. The protocol of this study was approved by the CEIm Parc de Salut Mar (CEIC: 2019/8622/I). This study respects the Code of Good Scientific Practices (http://www.imim.es/imim/cas/c-CBPC.htm).

Data confidentiality: The data included are pseudonymized and identified by an internal code generated upon entering them into the REDCAP database, which prevents the identification of the subjects included. Only the clinical investigators of the study have access to the patient identification data. The database will only contain the internal code. The project data will be stored in REDCAP at the institution.

The Hospital Mar Research Institute of Barcelona is responsible for the overall archive. The Hospital Mar Research Institute and the principal investigator of the project are responsible for data processing within the framework of this study. No copies will be made outside the REDCAP server.

In all the steps of the study, the confidentiality of the included subjects is guaranteed, in accordance with the provisions of Organic Law 3/2018, of December 5, on the protection of personal data and the guarantee of digital rights (LOPD 3/2018).

The study is subject to the ethical standards of biomedical research in humans. Patients are informed and given an informed consent document (in Spanish or Catalan as appropriate) approved by the local ethics committee. Aspects related to confidentiality follows established regulations. Participants are informed, in accordance with Organic Law

3/2018, that their data may be subject to automated processing and that they have the right to consult, modify, or delete their personal data file. They are informed that the database is managed by the principal investigator and the project researchers using an access code, and that the patient's name and medical history will not appear in the registry.

## Discussion

Frailty is, and will continue to be, one of the major challenges faced by our society, due to the growing ageing population and its profound implications for health systems and social care policies [15–17].

Although recent years have seen advances in the understanding and management of frailty, there is still limited knowledge regarding the events and mechanisms that drive the transition from robustness to frailty and, ultimately, to dependence.

From a scientific perspective, this project aims to generate valuable insights into the clinical and biological phenotypic clusters involved in the different stages of frailty. Additionally, it seeks to identify biological markers associated with frailty progression, allowing for earlier identification of at-risk individuals and monitoring of those prone to rapid deterioration. By uncovering these pathways, we hope to pinpoint potential therapeutic targets to improve patient management.

For this purpose, a prospective, multicenter, real-life cohort was designed to advance the understanding of frailty trajectories and their clinical impact. By combining clinical, functional, and biological data, the study aims to characterize phenotypic clusters of frailty and to elucidate mechanisms underlying disability progression. A key strength of the initiative is the validation of the FTS5 against established measures (Fried phenotype and eFI), assessing its feasibility and clinical applicability in both primary care and hospital settings. Furthermore, the creation of a biobank ensures long-term availability of biological samples for translational research.

The study's comprehensive methodology—including annual follow-up, multidimensional assessments, and integration of metabolomic and biochemical data—provides an unprecedented opportunity to analyze frailty as a dynamic process rather than a static condition. Advanced clustering and longitudinal modeling will allow the identification of distinct trajectories and their correlation with clinically meaningful outcomes such as healthcare utilization, hospitalization, and mortality.

Overall, this study represents a significant step forward in frailty research by bridging population-based epidemiology, clinical practice, and biological mechanisms. Its design is expected to contribute not only to the refinement of diagnostic tools but also to the development of tailored preventive and therapeutic strategies for older adults at risk of disability.

In summary, the project aims to advance our understanding of the interindividual variability in clinical trajectories that lead to frailty, dependence, and eventually, death. Specifically, this proposal will:

1. Enable early detection of individuals at risk of progressing to frailty and dependence by identifying characteristic phenotypic profiles.

2. Facilitate longitudinal monitoring of frailty progression and vulnerability in individual patients based on defined risk pathways.

3. Support the definition of standardized, well-characterized populations in which preventive or therapeutic interventions can be tested.

4. Provide a framework for the identification of homogeneous clinical clusters suitable for clinical and epidemiological research.

5. Lay the groundwork for future studies on biological markers related to the progression of frailty and vulnerability.

The overarching goal of the project is to minimize the impact of frailty and its consequences—particularly dependence—by improving diagnostic classification, prognostic evaluation, and the development of tailored management strategies for distinct risk phenotypes.

The results will be continuously disseminated to the scientific and medical community via meetings with other researchers, clinical sessions, and patient associations. Upon completion of the study, the findings will be published in peer-reviewed journals.

## Gantt chart

| Task name | Stard date | End date | Status | 2021 | 2022 | 2023 | 2024 | 2025 | 2026 | 2027 | 2028 | 2029 | 2030 |
|---|---|---|---|---|---|---|---|---|---|---|---|---|---|
| **PROJECT** | 02.01.2021 | 31.12.2030 | Open | x | x | x | x | x | x | x | x | x | x |
| Patient recruitment | 02.01.2021 | 31.12.2027 | In process | x | x | x | x | x | x | x | | | |
| Patient follow-up | 02.01.2021 | 31.12.2029 | In process | x | x | x | x | x | x | x | x | | |
| Creation of CIBERFES Biobank | 02.01.2021 | 31.12.2029 | In process | x | x | x | x | x | x | x | x | | |
| Sample analysis | 02.01.2028 | 31.12.2030 | pending | x | x | x | x | x | x | x | x | x | x |
| Data analysis | 02.01.2028 | 31.12.2030 | pending | x | x | x | x | x | x | x | x | x | x |
| *Report Results* | 2030 | | pending | | | | | | | | | | x |

### Last update of the study

Up to date, we have recruited 500 participants where 300 have a minimum of follow-up of one year.

## Supporting information

**S1 File. Original granted study protocol approved by Ethics Committee.**
(PDF)

**S1 Table. SPIROS checklist.**
(PDF)

## Author contributions

**Conceptualization:** Jose Antonio Carnicero Carreño, Leocadio Rodriguez Mañas, Xavier Nogués.

**Data curation:** Natalia Garcia-Giralt, Diana Ovejero, Jose Antonio Carnicero Carreño, Anna Ribes, Montse Rabassa.

**Formal analysis:** Natalia Garcia-Giralt, Diana Ovejero, Montse Rabassa, Àlex Sanchez Pla.

**Funding acquisition:** Natalia Garcia-Giralt, Xavier Nogués.

**Investigation:** Natalia Garcia-Giralt, Diana Ovejero, Jose Antonio Carnicero Carreño, Anna Ribes, Pedro Abizanda Soler, Jose Antonio Serra Rexach, Francisco José García García, Montse Rabassa, Leocadio Rodriguez Mañas, Àlex Sanchez Pla, Maria Cristina Andrés Lacueva, Xavier Nogués.

**Methodology:** Natalia Garcia-Giralt, Diana Ovejero, Jose Antonio Carnicero Carreño, Anna Ribes, Mariam El Assar de la Fuente, Carmen Maria Osuna Del pozo, Inmaculada Carmona, María Ángeles Caballero-Mora, Virginia Mazoteras Muñoz, Elisa Belen Cortes Zamora, Almudena Avendaño Céspedes, Bárbara Agud Andreu, Fernando Gómez Galera, Jade Soldado-Folgado.

**Project administration:** Natalia Garcia-Giralt.

**Supervision:** Jose Antonio Carnicero Carreño, Pedro Abizanda Soler, Leocadio Rodriguez Mañas, Maria Cristina Andrés Lacueva, Xavier Nogués.

**Validation:** Natalia Garcia-Giralt, Jose Antonio Serra Rexach, Francisco José García García, Montse Rabassa, Àlex Sanchez Pla, Xavier Nogués.

**Writing – original draft:** Natalia Garcia-Giralt, Montse Rabassa.

**Writing – review & editing:** Natalia Garcia-Giralt, Diana Ovejero, Jose Antonio Carnicero Carreño, Anna Ribes, Pedro Abizanda Soler, Jose Antonio Serra Rexach, Francisco José García García, Montse Rabassa, Leocadio Rodriguez

Mañas, Àlex Sanchez Pla, Mariam El Assar de la Fuente, Carmen Maria Osuna Del pozo, Inmaculada Carmona, María Ángeles Caballero-Mora, Virginia Mazoteras Muñoz, Elisa Belen Cortes Zamora, Almudena Avendaño Céspedes, Bárbara Agud Andreu, Fernando Gómez Galera, Jade Soldado-Folgado, Maria Cristina Andrés Lacueva, Xavier Nogués.

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
