## [Decision Letter · Decision Letter 0]

20 Jan 2026

Dear Dr.  Garcia-Giralt,

plosone@plos.org . A letter that responds to each point raised by the academic editor and reviewer(s). You should upload this letter as a separate file labeled 'Response to Reviewers'.A marked-up copy of your manuscript that highlights changes made to the original version. You should upload this as a separate file labeled 'Revised Manuscript with Track Changes'.An unmarked version of your revised paper without tracked changes. You should upload this as a separate file labeled 'Manuscript'.

We look forward to receiving your revised manuscript.

Kind regards,

Marta Ingles

Academic Editor

PLOS One

Journal Requirements:

“The present study is funded by CIBER on Frailty and Healthy Ageing, by Francisco Soria Melguizo Fundation, and FIS (ISCIII) num. PI19/00033. Additional support was provided by AGAUR (2021 SGR 00043), and FEDER, EU.”

4. Please note that funding information should not appear in any section or other areas of your manuscript. We will only publish funding information present in the Funding Statement section of the online submission form. Please remove any funding-related text from the manuscript.

6. Please include captions for your Supporting Information files at the end of your manuscript, and update any in-text citations to match accordingly. Please see our Supporting Information guidelines for more information: http://journals.plos.org/plosone/s/supporting-information .

7. Thank you for providing your underlying data as Supporting Information.

We note that the data set contains text or data that is not in English. Please note that PLOS is an English-language publisher, so we require data sets to be provided in English as well. Please upload an English-language version of your data set.

This will also allow us to determine if your data follows PLOS standards per our Data Availability policy here: https://journals.plos.org/plosone/s/data-availability

Additional Editor Comments:

Please note that one of the reviewers has recommended rejection, so please carefully consider and address all the points raised in their reports.

We have received comments from two reviewers who recommend revisions to the manuscript. We kindly ask that you carefully consider and address all the points raised in their reports. Please revise the manuscript accordingly and provide a detailed, point-by-point response explaining how each comment has been addressed or, where applicable, justify any suggestions that have not been incorporated.

Reviewer's Responses to Questions

**Comments to the Author**

1. Does the manuscript provide a valid rationale for the proposed study, with clearly identified and justified research questions?

Reviewer #1: Yes

Reviewer #2: Yes

2. Is the protocol technically sound and planned in a manner that will lead to a meaningful outcome and allow testing the stated hypotheses?

Reviewer #1: Yes

Reviewer #2: Yes

3. Is the methodology feasible and described in sufficient detail to allow the work to be replicable?

Reviewer #1: Yes

Reviewer #2: Yes

4. Have the authors described where all data underlying the findings will be made available when the study is complete?

Reviewer #1: No

Reviewer #2: Yes

5. Is the manuscript presented in an intelligible fashion and written in standard English?

Reviewer #1: Yes

Reviewer #2: Yes

You may also provide optional suggestions and comments to authors that they might find helpful in planning their study.

Reviewer #1: This is a study protocol to conduct a cohort study on frailty phenotyping. A CIBERFES Biobank will be created to store blood and urine samples from participants. Demographic and clinical

history data, anthropometric measurements, the PREDIMED questionnaire, peripheral

blood biochemical variables, and metabolomics data will be collected at baseline and

annually until participants develop frailty. Statistical analysis includes clustering patients into groups with similar deficits and characteristics.

I only have some minor issues for authors to improve the presentation.

Terminology Corrections

• In Section 3.4, replace “alpha risk” with “Type I error”, and “beta risk” with “Type II error”, to align with the standard terminology used in medical and clinical studies.

• Replace “bilateral” with “two-sided” when referring to hypothesis tests, as this is the conventional phrasing in statistics.

• Revise “a factor of risk with a minimum RR” to “a minimum relative risk value” for clarity and precision.

Clarification Needed

• Please clarify what you mean by “POISSON approach”. Are you referring to:

o a Poisson regression model for count data?

o a Poisson distributional assumption used for approximating rare events?

o or a specific Poisson-based test procedure?

It is best to avoid using so many bulleted items in presentations, especially for many short terms and phrases. Including them in a formal paragraph would be more human-like writing style. Overuse of bullet points, especially for short terms or simple phrases, can make a presentation feel fragmented or mechanical.

Section 3.7, so do you expect the two clusters to be fast and slow groups? Will they differ in onset and accumulation of deficits? Change the sentence “Results will be used …” to make your point clear.

What do you mean “frailty … will be correlated with …”? Do you mean you want to establish some regression models using these variables? Which variable is response? Then you need to describe some more details about the regression analysis to be carried out. You probably will also employ some significance tests at some point along the development.

It is also strange to use the data to build the cluster (in a unsupervised manner since cluster status unknown) and then use the same data to build a regression model to predict the identified cluster. I don’t think this is a good plan.

Reviewer #2: The proposal was very detailed, addressing a critical aspect of health. The collection of bio samples contributes to the significance of the study and distinguishes it from previous studies. However, there were some grammatical errors in the paper which should be revised before acceptance. For example in the review of previous research, the author states "this is crucial because frailty is not an inevitable consequence of aging but is potentially reversible, even spontaneously, especially in its early stages". It is the assumption that this sentence was included to affirm the need for early frailty diagnosis, but the intent is not very clear in the sentence. I would recommend rephrasing that sentence to convey the message clearly.

There were also some statements that were not very clear in the methods. According to the author, the participants would be required to complete at least 3 visits out of 4 (one of which includes the recruitment and signing of informed consent). This suggests that there is a possibility that participants would visit the hospital for the sole purpose of participating in the study, but based on the sampling method described in the protocol, the authors intend to use a convenience sampling from the patient population at the included study sites.

Finally, given that the population are likely Spanish speakers, the authors did not describe any plans for translation of the study tools. They also did not indicate the language the informed consent would be given in.

Overall, this is a relevant study.

**Do you want your identity to be public for this peer review?** For information about this choice, including consent withdrawal, please see our Privacy Policy

Reviewer #1: No

Reviewer #2: No

---

## [Author Response · Author response to Decision Letter 1]

11 Feb 2026

Response to reviewers

Dear Editor,

We are grateful for the opportunity to amend our manuscript. We have carefully reviewed the comments and have revised the manuscript accordingly. The revisions are included in the attached new version of the manuscript; all new sentences and other changes are shown using track changes.

Our responses are provided in a point-by-point manner below.

We truly appreciate all the constructive comments and suggestions from reviewers that

have improved our manuscript.

Responses

1)Comments to the Author

Have the authors described where all data underlying the findings will be made available when the study is complete?

Reviewer #1: No ------------A data availability statement has been included in the protocol

Reviewer #2: Yes

2. Review Comments to the Author

Reviewer #1: This is a study protocol to conduct a cohort study on frailty phenotyping. A CIBERFES Biobank will be created to store blood and urine samples from participants. Demographic and clinical

history data, anthropometric measurements, the PREDIMED questionnaire, peripheral

blood biochemical variables, and metabolomics data will be collected at baseline and

annually until participants develop frailty. Statistical analysis includes clustering patients into groups with similar deficits and characteristics.

I only have some minor issues for authors to improve the presentation.

1) Terminology Corrections

• In Section 3.4, replace “alpha risk” with “Type I error”, and “beta risk” with “Type II error”, to align with the standard terminology used in medical and clinical studies.

• Replace “bilateral” with “two-sided” when referring to hypothesis tests, as this is the conventional phrasing in statistics.

• Revise “a factor of risk with a minimum RR” to “a minimum relative risk value” for clarity and precision.

Response: Thanks to point it out. We have amended the text.

2) Clarification Needed

• Please clarify what you mean by “POISSON approach”. Are you referring to:

o a Poisson regression model for count data?

o a Poisson distributional assumption used for approximating rare events?

o or a specific Poisson-based test procedure?

Response: Thank you for the suggestion; We have used a Poisson distribution since the primary outcome is a count of events occurring over a fixed period of time (or per unit of exposure), especially it can be useful if events are rare and independent.

We have amended the text: “sample size was calculated using a Poisson distribution approach, appropriate for the analysis of event rates. The calculation was based on the expected incidence rate of the outcome, the anticipated rate ratio between groups, a two sided significance level of 0.05, and a statistical power of 80%.”

3) It is best to avoid using so many bulleted items in presentations, especially for many short terms and phrases. Including them in a formal paragraph would be more human-like writing style. Overuse of bullet points, especially for short terms or simple phrases, can make a presentation feel fragmented or mechanical.

Response: thank you for the suggestion. We have made efforts to reduce the number of bullet points. However, given that this is a study protocol, we think that certain variables are better understood when presented in a schematic format.

4) Section 3.7, so do you expect the two clusters to be fast and slow groups? Will they differ in onset and accumulation of deficits? Change the sentence “Results will be used …” to make your point clear.

What do you mean “frailty … will be correlated with …”? Do you mean you want to establish some regression models using these variables? Which variable is response? Then you need to describe some more details about the regression analysis to be carried out. You probably will also employ some significance tests at some point along the development.

It is also strange to use the data to build the cluster (in a unsupervised manner since cluster status unknown) and then use the same data to build a regression model to predict the identified cluster. I don’t think this is a good plan.

Response: We agree that the text was overly summarized and that some aspects of the analyses may have been misunderstood. We have therefore expanded the explanation to improve clarity and provide a more detailed description of the statistical analysis plan.

“To identification of frailty phenotypes, unsupervised cluster partitioning methods will be applied to identify groups of individuals with similar patterns of deficits and clinical characteristics, thereby defining distinct frailty phenotypes. Specifically, k means clustering and hierarchical clustering approaches will be used. Prior to clustering, variables will be standardized as appropriate to ensure comparability across measures. The optimal number of clusters and the quality of the clustering solutions will be evaluated using predefined validation criteria, including the gap statistic and silhouette index. These metrics will be used to assess cluster separation, cohesion, and stability, as well as the robustness of individual cluster assignments.

In a second step, a longitudinal follow-up will be conducted to record the onset and accumulation of health deficits throughout the study period, continuing until the end of follow‑up, death, or loss to follow‑up. To delineate distinct patterns of frailty progression over time, we will apply latent class analysis using mixed growth models or comparable latent trajectory modelling approaches. These methods will allow to identify differential longitudinal trajectories characterized either by varying rates of deficit accumulation (capturing acceleration in frailty development) or by distinct combinations of specific deficits or phenotypic characteristics. The resulting models will classify participants into latent clusters representing heterogeneous frailty trajectories, enabling the identification of subgroups at higher risk of accelerated decline and facilitating targeted intervention strategies.

Finally, the identified frailty phenotypes and patient clusters will be examined in relation to clinically relevant health outcomes, including the utilization of health care services in both primary and secondary care settings, hospital admissions, biomarker profiles, and mortality. Appropriate regression models will be used depending on the outcome of interest, and results will be reported using effect estimates with corresponding 95% confidence intervals. Where applicable, analyses will account for follow up time and relevant covariates. These analyses will enable the assessment of the prognostic and clinical relevance of the identified frailty phenotypes and trajectories.”

Reviewer #2: The proposal was very detailed, addressing a critical aspect of health. The collection of bio samples contributes to the significance of the study and distinguishes it from previous studies. However, there were some grammatical errors in the paper which should be revised before acceptance.

1)For example in the review of previous research, the author states "this is crucial because frailty is not an inevitable consequence of aging but is potentially reversible, even spontaneously, especially in its early stages". It is the assumption that this sentence was included to affirm the need for early frailty diagnosis, but the intent is not very clear in the sentence. I would recommend rephrasing that sentence to convey the message clearly.

Response: Thank you for the suggestion. We have changed the sentence to “This aspect is particularly important because frailty is not an inevitable consequence of aging; instead, it represents a dynamic and potentially reversible condition, with early stages offering the greatest opportunity for recovery.” Moreover, we have revised all the text.

2) There were also some statements that were not very clear in the methods. According to the author, the participants would be required to complete at least 3 visits out of 4 (one of which includes the recruitment and signing of informed consent). This suggests that there is a possibility that participants would visit the hospital for the sole purpose of participating in the study, but based on the sampling method described in the protocol, the authors intend to use a convenience sampling from the patient population at the included study sites.

Response: We fully agree that we did not express this point clearly in the initial version. For this reason, we have rewritten the text to ensure greater clarity.

“The recruitment process comprises a preselection visit to assess inclusion and exclusion criteria and to inform the participant about the study, followed by a baseline visit (V0) during which all study variables will be collected and written informed consent will be obtained. Each participant will complete a minimum of three study visits. These will include the baseline visit (V0) and 2 years of follow-up (V1 at 1 year of follow-up and V2 at two years of follow-up). Follow-up visits will be conducted every 12 months until the end of the study;”

3) Finally, given that the population are likely Spanish speakers, the authors did not describe any plans for translation of the study tools. They also did not indicate the language the informed consent would be given in.

Response: Thank you for your comment. The original informed consent form is indeed in Spanish or Catalan (the language of Catalonia), and we have translated it into English to meet the journal's submission requirements. We have included this statement in the Ethical section.

---

## [Editor Report · Decision Letter 1]

3 Mar 2026

Study protocol to establish a prospective cohort for the study of phenotypic clusters, progression pathways, and outcomes of frailty and dependence: The CohorFES

PONE-D-25-37732R1

Dear Dr. Garcia-Giralt,

We’re pleased to inform you that your manuscript has been judged scientifically suitable for publication and will be formally accepted for publication once it meets all outstanding technical requirements.

Kind regards,

Marta Ingles

Academic Editor

PLOS One

Additional Editor Comments (optional):

I consider that the authors have responded satisfactorily to all the reviewers’ comments and revisions, and I therefore recommend that the manuscript be accepted for publication.
---

## [Editor Report · Acceptance letter]

PONE-D-25-37732R1

PLOS One

Dear Dr. Garcia-Giralt,

I'm pleased to inform you that your manuscript has been deemed suitable for publication in PLOS One. Congratulations! Your manuscript is now being handed over to our production team.

Kind regards,

on behalf of

Dr. Marta Ingles

Academic Editor

PLOS One